# Identification of Molecules from Coffee Silverskin That Suppresses Myostatin Activity and Improves Muscle Mass and Strength in Mice

**DOI:** 10.3390/molecules26092676

**Published:** 2021-05-03

**Authors:** Jeong Han Kim, Jae Hong Kim, Jun-Pil Jang, Jae-Hyuk Jang, Deuk-Hee Jin, Yong Soo Kim, Hyung-Joo Jin

**Affiliations:** 1Department of Marine Molecular Bioscience, Gangneung-Wonju National University, Gangneung-si 25457, Korea; sinan1708@naver.com (J.H.K.); kej1345@naver.com (J.H.K.); dhjin@gwnu.ac.kr (D.-H.J.); 2Anticancer Agent Research Center, Korea Research Institute of Bioscience and Biotechnology, Cheongju 28116, Korea; jpjang@kribb.re.kr; 3Natural Medicine Research Center, Korea Research Institute of Bioscience and Biotechnology, Cheongju 28116, Korea; jangjh@kribb.re.kr; 4Department of Human Nutrition, Food and Animal Sciences, University of Hawaii, 1955 East-West Rd., Honolulu, HI 96822, USA

**Keywords:** coffee silverskin, myostatin inhibition, muscle atrophy, *^β^N*-arachinoyl−5-hydroxytryptamide, *^β^N*-behenoyl−5-hydroxytryptamid

## Abstract

Coffee has been shown to attenuate sarcopenia, the age-associated muscle atrophy. Myostatin (MSTN), a member of the TGF-β growth/differentiation factor superfamily, is a potent negative regulator of skeletal muscle mass, and MSTN-inhibition increases muscle mass or prevents muscle atrophy. This study, thus, investigated the presence of MSTN-inhibitory capacity in coffee extracts. The ethanol-extract of coffee silverskin (CSE) but not other extracts demonstrated anti-MSTN activity in a pGL3-(CAGA)_12_-luciferase reporter gene assay. CSE also blocked Smad3 phosphorylation induced by MSTN but not by GDF11 or Activin A in Western blot analysis, demonstrating its capacity to block the binding of MSTN to its receptor. Oral administration of CSE significantly increased forelimb muscle mass and grip strength in mice. Using solvent partitioning, solid-phase chromatography, and reverse-phase HPLC, two peaks having MSTN-inhibitory capacity were purified from CSE. The two peaks were identified as *^β^N*-arachinoyl−5-hydroxytryptamide (C_20_−5HT) and *^β^N*-behenoyl−5-hydroxytryptamide (C_22_−5HT) using mass spectrometry and NMR analysis. In summary, the results show that CSE has the MSTN-inhibitory capacity, and C_20_−5HT and C_22_−5HT are active components of CSE-suppressing MSTN activity, suggesting the potential of CSE, C_20_−5HT, and C_22_−5HT being developed as agents to combat muscle atrophy and metabolic syndrome.

## 1. Introduction

Myostatin (MSTN), also known as growth and differentiation factor-8 (GDF8), is mainly expressed in skeletal muscle and acts as a negative regulator of skeletal muscle growth in animals [1]. During embryonic development, MSTN suppresses muscle cell hyperplasia by inhibiting the proliferation of muscle progenitors and myoblasts [2,3]. MSTN also plays an important role in the growth and maintenance of adult skeletal muscle mass [4,5]. The upregulation of MSTN is closely related to muscle atrophy caused by various physiological conditions and illnesses [6,7,8,9]. Many studies have shown that suppressing MSTN activity using various approaches, such as administration of MSTN-blocking proteins and peptides, delivery of MSTN-blocking genes, and RNAi method increases muscle mass in adult animals [10]. Studies have also demonstrated that MSTN-blocking is a potential therapeutic strategy to combat muscle wasting occurring in a wide range of serious illness and age-associated sarcopenia in animal models [11,12], as well as in human clinical trials [13,14,15,16]. In addition, MSTN suppression improved the insulin sensitivity and whole-body metabolism accompanied by a decrease in fat mass in mice and pigs [17,18,19,20], indicating that MSTN plays a role not only in the regulation of muscle mass but, also, in the regulation of fat mass and energy metabolism.

There has been increasing interest in natural compounds from commonly consumed dietary or edible plants as a source of drug candidates for muscle atrophy, and extracts, fractions, or compounds of various dietary plants have demonstrated their potential for the treatment of muscle atrophy [21]. Given that MSTN is a strong negative regulator of skeletal muscle growth and is upregulated in muscle atrophic conditions, examining the MSTN-inhibitory capacities of the extracts and fractions of potential dietary plants appears to be a strategy in searching for natural compounds to treat muscle atrophy. A recent study showed that quercetin, a flavonoid widely distributed in edible plants, has a protective effect on DEX-induced muscle atrophy in a muscle cell culture model [22]. The same study also observed that quercetin enhanced the activation of Akt, a downstream effector of the MSTN signaling pathway, suggesting a potential capacity of quercetin to suppress MSTN activity. Coffee has been reported to attenuate the progression of sarcopenia and to increase the regenerating capacity of injured muscles in aged mice [23]. A decrease in blood proinflammatory cytokines and an increase in Akt activation were observed in coffee-treated mice as compared with control mice. We thus hypothesized that the coffee bin contains compounds capable of MSTN inhibition.

This study was designed to identify compounds that are capable of suppressing the activity of MSTN from coffee bin extract. Our results show that ethanol extracts of coffee silverskin (CSE) suppressed MSTN activity, and its administration to mice increased muscle mass and grip strength along with a decrease in blood fatty acid concentration. *^β^N*-arachinoyl-5-hydroxytryptamide (C_20_-5HT) and *^β^N*-behenoyl-5-hydroxytryptamide (C_22_-5HT) were identified as the active constituents of CSE suppressing MSTN activity.

## 2. Results

### 2.1. Coffee Silverskin Ethanol-Extract (CSE) Suppresses MSTN Activity

Using the Colombia supremo huila variety, ethanol- and water-extracts of green beans, roasted beans, and silverskin were prepared to determine the presence of anti-MSTN substances in those extracts. The ethanol extract of silverskin (CSE) showed anti-MSTN activity, but other extracts showed little MSTN-inhibitory activity (Figure 1a). Since only CSE showed MSTN-inhibitory activity, the CSE of eight coffee varieties, including Colombia supremo huila, India gangagiri, Ethiopia yirgacheffee, Guatemala SHB huehuetenango, Kenya AA FAQ, Kenya AB TOP, Brazil NY2FC17/18 Cerrado, and Ethiopia kochere onacho, were prepared, and their capacities to suppress MSTN activity were examined. The IC_50_ values of the above eight varieties to suppress MSTN activity were 0.60, 2.29, 2.78, 0.78, 1.31, 0.78, 0.62, and 0.80 µg/mL, respectively (Figure 1b). Colombia supremo huila, Guatemala SHB huehuetenango, Kenya AB TOP, Brazil NY2FC17/18 Cerrado, and Ethiopia kochere onacho showed significantly stronger anti-MSTN activity than Kenya AA FAQ. India gangagiri and Ethiopia yirgacheffee showed the lowest MSTN-inhibitory capacity.

The capacity of CSE mixture of various varieties to suppress MSTN activity was compared to its capacity to suppress GDF11 and Activin A, the molecules closely related to MSTN, using the luciferase reporter assay. The IC_50_ for MSTN was 0.79 µg/mL, but the IC_50_ for GDF11 and Activin A were not measurable over the concentration range tested (Figure 2), indicating that CSE’s suppression of MSTN is more specific than its suppression of Activin A or GDF11.

Since CSE demonstrated MSTN inhibitory capacity, we examined whether CSE inhibits MSTN binding to its receptor. MSTN signals through binding predominantly to activin type IIB (ActRIIB) and two type I receptors (ALK4/5) to elicit its biological function [24,25]. The binding of MSTN to ActRIIB recruits and activates type I and type IIB activin receptors, resulting in the phosphorylation of Smad2/3 transcription factors to regulate the expression of downstream target genes, as well as the crosstalk with other signaling pathways [26,27,28]. GDF11 and Activin A are also known to have signaling pathways similar to MSTN, binding to ActRIIB and activating Smad2/3 pathways [29,30]. Thus, the phosphorylation of Smad3 was examined in HepG2 cells treated by MSTN, GDF11, Activin A, with or without CSE. Smad3 was phosphorylated in cells treated by MSTN, and the phosphorylation was suppressed when treated by MSTN with CSE (Figure 3). Both GDF11 and Activin A phosphorylated Smad3 (Figure 3), indicating that MSTN, GDF11, and Activin A have similar signaling pathways. In contrast to MSTN, the phosphorylation of Smad3 induced by GDF11 or Activin A was not inhibited by CSE (Figure 3), demonstrating that CSE inhibits the Smad signaling pathway induced by MSTN but not induced by GDF11 or Activin A. Given that MSTN, GDF-11, and Activin A share the same receptor for their signaling, it appears that the CSE inhibition of Smad3 phosphorylation is via CSE binding to MSTN but not via CSE binding to its receptor, ActRIIB.

### 2.2. Oral Administration of CSE Increases the Muscle Mass and Grip Strength but Not Body Weight and Bone Mass in Mice

Since CSE showed MSTN-inhibitory activity in vitro, we examined in vivo effects of CSE in mice. The oral administration of CSE for 29 days resulted in a significant increase in the forelimb’s muscle mass (2.28 ± 0.142 vs. 1.70 ± 0.102 mg, *p* < 0.001) as compared with the control group, but the hindlimb’s muscle mass was not significantly different from the control group (2.68 ± 0.231 vs. 2.60 ± 0.101 mg) (Figure 4c). Additionally, the grip strength of CSE-treated mice was significantly greater as compared with that of the control mice on day 22 (0.21 ± 0.079 vs. 0.11 ± 0.068 N, *p* < 0.05) and on day 29 (0.31 ± 0.161 vs. 0.09 ± 0.073 N, *p* < 0.05) but not on day 8 and 15 (Figure 4b). Body and bone weights were not affected by the CSE administration (Figure 4a,d).

### 2.3. CSE Administration Decreases the Blood Free Fatty Acid Level

The free fatty acid level in the blood of CSE-treated mice tended to be lower than that of the control group (588.6 ± 127.37 µEq/L vs. 745.8 ± 119.59 µEq/L, *p* < 0.1). However, glucose, total cholesterol, and triglyceride in the blood between the two groups were not significantly different (Table 1).

### 2.4. Administration of CSE Upregulated the mRNA Expression Level of Ppargc1a and Ucp1 in Mice

The effect of the oral administration of CSE on the expression of mRNA genes related to energy metabolism was examined. The mRNA expression levels of the Ppargc1a (143%, *p* < 0.05) and Ucp1 (508%, *p* < 0.01) genes were significantly upregulated in the CSE-treated group compared with the control group, while the mRNA expression levels of Fndc5 and Mstn were not significantly different between the two groups (Figure 5).

### 2.5. Purification and Structural Identification of Compounds with MSTN-Inhibitory Capacity

Solvent partitioning results showed that chloroform and hexane fractions inhibited more than 80% of 1-nM MSTN activity while water and 60% ethanol fractions inhibited less than 20% of 1-nM MSTN activity in HEK293 cells carrying (CAGA)_12_-luciferase gene construct (Figure 6a). Solid-phase extraction chromatography of the chloroform fraction with the C18 Sep-Pak cartridge showed that 80% and 100% methanol fractions had MSTN-inhibitory capacities (50% and 70%, respectively) in HEK293 cells carrying (CAGA)_12_-luciferase gene construct (Figure 6b). 

Reversed-phase HPLC of the 100% methanol fraction showed two peaks (Fractions 16 and 19) with MSTN-inhibitory capacity (Figure 7a,b), and the inhibitions were dose-dependent (Figure 7c).

In the electrospray ionization mass spectrometry spectrum (ESIMS), peak1 (fraction 15) showed a protonated [M + H]^+^ ion at m/z 471.4. The ^1^H NMR spectrum of peak1 evidenced the presence of a long alkyl chain due to characteristic signals (δ_H_ 0.92 and 1.30), three aromatic protons (δ_H_ 7.16, 6.95, and 6.67), an olefinic proton (δ_H_ 7.02), and two methylene protons (δ_H_ 3.46 and 2.88). The UV spectrum of peak1 was similar to those of 5-hydroxytryptamine with the UV absorption patterns at 280 and 300 nm. Additionally, the spectroscopic data comparison between observed and previously reported data further confirmed that the peak1 is ^β^N-arachinoyl-5-hydroxytryptamide (C_20_-5-HT) (Figure 8a and Appendix A). The NMR data of the peak2 were similar to those of ^β^N-arachinoyl-5-hydroxytryptamide, but the ESIMS showed a molecular ion peak at m/z 499 ([M]^+^), which exceeded that of the latter compound by 28 amu. Additionally, the ^13^C NMR spectrum of peak2 gave additional methylene signals at δ_C_ 28.4−29.7, in comparison with that of ^β^N-arachinoyl-5-hydroxytryptamide. Thus, the structure of the peak2 was elucidated as ^β^N-behenoyl-5-hydroxytryptamide (C_22_-5-HT) (Figure 8b and Appendix A).

## 3. Discussion

This study first introduces a novel function of the ethanol extract of coffee silverskin (CSE) by showing that CSE suppressed the activity of MSTN, a potent negative regulator of skeletal muscle mass. The MSTN inhibitory capacity of CSE measured by pGL-(CAGA)_12_ luciferase reporter gene assay was further confirmed by CSE’s blocking of Smad3 phosphorylation, a critical component of the canonical MSTN signaling pathway. MSTN, as a member of the TGF-β superfamily, is closely related to Activin A and GDF11 and shares functional redundancy and signaling process via Smad phosphorylation with these ligands, but at the same time, these ligands elicit their specific cellular effects [31]. In this regard, MSTN-inhibitors need to be specific for MSTN inhibition with minimal inhibition of other TGF-β superfamily ligands to minimize the nontarget effects in vivo. The current result shows that CSE suppressed MSTN in a pGL-(CAGA)_12_ luciferase reporter gene assay, while Activin A and GDF11 were not inhibited by CSE in the concentration range tested. Furthermore, CSE blocked Smad3 phosphorylation induced by MSTN but not by GDF11 or Activin A, indicating that CSE has the potential as a specific MSTN inhibitor.

Studies have shown that the suppression of MSTN activity increases muscle mass in adult animals [10], as well as improves body metabolism and insulin sensitivity [17,20]. Since CSE demonstrated MSTN-inhibitory capacity in vitro experiment, we examined the effects of CSE on muscle mass and strength and blood metabolites by daily oral administration of CSE for 29 days in mice. The administration of CSE significantly increased forelimb muscle mass with a concomitant increase in forelimb grip strength. Unexpectedly, hindlimb muscle mass was not affected by the CSE administration. With the current data, the reason for the differential effect cannot be explained, and future studies are needed to confirm and explain the result with detailed mechanistic investigations. Notably, despite the increase in forelimb muscle mass, the body weight was not affected by the administration of CSE. The fat content was not measured in this experiment. Given that MSTN inhibition is known to reduce fat accumulation in mice [17,18], it is likely that the reason for no change in body weight was due to a decrease in fat mass. The free fatty acid concentration in serum was tended to decrease by the administration of CSE, indicating an active utilization of blood free fatty acids by CSE administration. This result is in agreement with other studies that also reported a decrease in blood free fatty acid level by the administration of MSTN-inhibitory compounds in mice fed either a regular [32] or high-fat diet [33]. However, the levels of blood glucose and triglyceride were not affected by the administration of CSE in contrast to the results of the above reports [17,31], where decreases in blood glucose and triglyceride were observed by MSTN-inhibitory compound administration. Aligning with the enhanced use of fatty acid by MSTN-inhibition, a study showed that MSTN-inhibition increased the expression of genes regulating lipid metabolism and energy expenditure, such as the *Ppargc1a**, Ucp1,* and *Fndc5* genes [17]. Ppargc1a upregulates genes involved in the TCA and mitochondrial fatty acid cycles and mitochondrial biogenesis to regulate energy metabolism [34,35,36]. Both the *Ucp−1* and *Fndc5* genes, adipose browning markers, have shown to be upregulated by the *Ppargc1a* [37,38]. Our results also showed that the *Ppargc1a and Ucp1* gene expression was upregulated by CSE administration, while the *Fndc5* gene expression was not significantly upregulated, even though the mean was greater (122%, *p* = 0.129). These results, the increase in muscle mass and strength, suppressed level of blood fatty acids, and enhanced expression of genes in lipid metabolism, suggest that the CSE administration suppressed MSTN activity in vivo. Furthermore, the results also suggest that the MSTN-inhibiting compound(s) present in CSE is orally active in suppressing MSTN activity in vivo. This study, however, could not thoroughly examine underlying signaling pathways associated with muscle protein metabolism regulated by MSTN, as well as muscle cellularities, such as muscle fiber hypertrophy and hyperplasia, and fiber-type composition, thus future studies need to investigate the underlying mechanisms.

Our results indicated that the potential compound(s) suppressing MSTN activity is mostly present in silverskin and is ethanol-soluble, since ethanol extracts of green beans and roasted beans, as well as the water extract of silverskin, showed little MSTN-inhibitory activity. Caffeine and chlorogenic acid, which are most abundant in green and roasted coffee beans, have various biological activities [34,39]. Given that ethanol- or water-extract of green or roasted coffee beans showed little MSTN-inhibitory activity, it is unlikely that caffeine or chlorogenic acid contributed to the MSTN-inhibitory capacity. Our purification and identification results showed that *^β^N*-arachinoyl-5-hydroxytryptamide (C_20_-5-HT) and *^β^N*-behenoyl-5-hydroxytryptamide (C_22_-5-HT) isolated from CSE inhibit MSTN activity. The N-alkanoyl-5-hydroxytryptamides, including C_20_-5-HT and C_22_-5-HT, are found in the waxy layer of green coffee beans as the major constituent [40,41], as well as in roasted coffee powder [42]. Of note, the ethanol-extract of green coffee bean which was expected to have N-alkanoyl-5-HT did not show MSTN-inhibitory capacity, suggesting that the concentration of C_20_-5-HT and/or C_22_-5-HT in the ethanol extract of the green coffee bean is considerably lower than the CSE.

In summary, results show that ethanol extract of coffee silverskin (CSE) suppresses MSTN activity in vitro, and its administration increased the muscle mass and strength, along with a decrease in the level of blood free fatty acids in mice. C_20_-5HT and C_22_-5HT were isolated from CSE as the major compounds contributing to the MSTN inhibition, suggesting the potential of CSE, C_20_-5HT, and C_22_-5HT being developed as agents to combat muscle atrophy and metabolic syndrome.

## 4. Materials and Methods

### 4.1. Coffee Beans

Green coffee beans, roasted coffee beans, and coffee silverskins of eight different varieties, including Colombia huila supremo, India gangagiri, Ethiopia yirgacheffee, Guatemala SHB huehuetenango, Kenya AA FAQ, Kenya AB TOP, Brazil NY2FC17/18 Cerrado, and Ethiopia kochere onacho, were obtained from the Haeram Coffee Company (Gangneung, Korea). The samples were ground to powder using a coffee grinder following the procedure described previously [43].

### 4.2. Preparation of Ethanol and Water Extracts of Samples

To extract the ethanol-soluble fraction of green coffee beans, coffee silverskins, and roasted coffee beans, 20-g powder samples were mixed with 1 L of 100% ethanol and incubated at room temperature for 24 h under a dark condition to reduce light-induced chemical reactions. Two additional extractions were performed and combined. To extract the water-soluble fraction, 20-g powder samples were boiled in 1 L of water for 20 min. The debris residue was removed by centrifugation (3000× *g*, 10 min). The weights of ethanol-soluble and water-soluble extracts were measured after complete evaporation of ethanol and water under vacuum at 30 °C and 50 °C, respectively. After the resolubilization of the ethanol and water extracts, the extracts were filtered through a 0.22-micron syringe filter before use, as described previously [43].

### 4.3. pGL3-(CAGA)_12_-Luciferase Assay

The luciferase assay was performed in the same manner as in the previous study [44]. Briefly, HEK293 cells stably expressing (CAGA)_12_-luciferase gene construct [45] were seeded on 96-well plates at 2 × 10^4^ cell//well and grown in DMEM containing 10% FBS, 1% penicillin/streptomycin, and 1% geneticin at 37 °C with 5% CO_2_ for 24 h. After incubation, the medium was replaced with 100-μL serum-free DMEM, then 1-nM MSTN, Activin A, or growth and differentiation factor 11 (GDF11) (R&D Systems, Minneapolis, MN, USA), plus various concentrations of the water or ethanol extracts of green bean, coffee silverskin, or roasted coffee bean were added to each well and incubated for 24 h. After incubation, luciferase activity was measured by a microplate luminometer (Berthold, Oak Ridge, TN, USA) using the Bright-Glo luciferase assay system (Promega, Madison, WI, USA). The percentage inhibition of MSTN activity was calculated by the following formula: Percentage of inhibition capacity = (luminescence at 1-nM ligands: MSTN, Activin A, or GDF11)—luminescence at the concentration of each extract) × 100/(luminescence at 1-nM ligands: MSTN, Activin A, or GDF11)—luminescence at 0-nM ligands (MSTN, Activin A, or GDF11). The IC_50_ (ligand concentration inhibiting 50% of MSTN, GDF11, or Activin A activity) values were estimated by a nonlinear regression model defining a dose-response curve using the Prism6 program (GraphPad, San Diego, CA, USA). The equation for the model was Y = Bottom + (Top-Bottom)/[1 + 10^(X-LogIC_50_)], where Y is % inhibition, Bottom is the lowest value of % inhibition set at 0%, Top is the highest value of % inhibition set at 100%, and X is Log ligand concentration. IC_50_ values were analyzed by ANOVA using the same program.

### 4.4. Western Blot Analysis

The effects of the coffee silverskin ethanol extracts (CSE; mixture of various varieties) on MSTN-, GDF11-, and Activin A-induced Smad3 phosphorylation were examined by the Western blot analysis following the procedure previously described [32]. HepG2 cells were seeded in 6-well plates at 2 × 10^5^ cells per well and grown in DMEM containing 10% FBS and 1% penicillin/streptomycin at 37 °C with 5% CO_2_ for 24 h. Cells were then switched to serum-free DMEM for 4 h, followed by treatment with 10-nM MSTN (R&D Systems, Minneapolis, MN, USA), 10-nM MSTN plus CSE (50 µg/mL), 10-nM GDF11 (R&D Systems, Minneapolis, MN, USA), 10-nM GDF11 plus CSE (50 µg /mL), 10-nM Activin A (R&D Systems, Minneapolis, MN, USA), and 10-nM Activin A plus CSE (50 µg /mL) for 30 min. Then, cells were lysed in RIPA buffer (20-mM Tris-HCl, pH 7.5, 150-mM NaCl, 1-mM EDTA, 1-mM EGTA, 1% NP-40, 1% sodium deoxycholate, 2.5-mM sodium pyrophosphate, 1-mM β-glycerophosphate, 1-mM Na_3_VO_4_, and 1-µg/mL leupeptin (Cell Signaling Technology, Danvers, MA, US) containing the protease and phosphatase inhibitor cocktails (Roche, Nutley, NJ, USA). The lysates were sonicated for 15 s in ice and centrifuged at 14,320× *g* for 20 min at 4 °C. Protein lysates were quantified by the BCA assay (Thermo Scientific, Waltham, MA, USA). Fifty micrograms of protein were separated by 10% SDS-PAGE and transferred to polyvinylidene fluoride membranes (PVDF; Millipore, Danvers, MA, USA). The membranes were blocked with 5% BSA in Tris-buffered saline (TBS) containing 0.1% Tween 20 (TTBS) for 3 h at room temperature. To assess phosphorylated and total Smad3, the membranes were incubated with primary antibodies specific for target proteins for 3 h at room temperature. The following primary monoclonal antibodies were purchased from Cell Signaling Technology: Smad3 (1:1000) and phospho-Smad3 (1:1000). The membranes were washed with TTBS for 10 min four times and incubated for 3 h at room temperature with horseradish peroxidase-conjugated anti-mouse IgG (1:5000). After washing, the reactive bands were detected by the SuperSignal West Femto Maximum Sensitivity Substrate (Thermo Scientific, Waltham, MA, USA) and visualized on Kodak Omat X-ray films. Densitometry analysis of the images obtained from X-ray films was performed using the Image J software (NIH).

### 4.5. Animal Experiment

#### 4.5.1. Animals

ICR male mice (6-week-old and weighing 30–35 g) were obtained from Central Lab Animal Inc. (Seoul, Korea). We followed the current regulations for the care and use of laboratory animals, and all the procedures of animal care were approved by the Animal Ethics Committee of Gangneung-Wonju National University, Gangneung, Korea (protocol #, GWNU-2017-19). The mice were maintained in a room at a constant temperature of 24 ± 1 °C under a 12-h light/12-h dark cycle at 65% humidity. Pellet diet (Purina, Seoul, Korea) and water were provided ad libitum. For a CSE (ethanol extract silverskin) oral administration experiment, mice (6-week-old and weighing 30–35 g) were randomly divided into two groups (*n* = 6 per group). CSE (mixture of various varieties) was diluted in 0.9% saline solution (0.75 µg/µL) and orally administered (200 µL) daily for 29 days in the treatment group. Mice in the control group were orally administered with 0.9% saline solution (200 µL). Body weight was measured on days 0, 8, 15, 22, and 29.

On day 29, after grip strength measurement, mice fasted for 12 h. After fasting, mice were anesthetized with an intramuscular injection of Zoletil 50 (Virbac, Carros, France; 10 mg/kg), and blood was collected by cardiac puncture. Serum was prepared by centrifugation at 890× *g* for 15 min at 4 °C and stored at −70 °C until use. After blood collection, the skin was peeled off; then, forelimbs and hindlimbs were disarticulated from scapula to carpus and from ilium to medial malleolus, respectively. The weight of forelimbs and hindlimbs were immediately measured, and the bones were removed immediately from the forelimb and hindlimb and weighed and frozen in liquid nitrogen, then stored at −70 °C until use. The muscle weight was calculated by subtracting the bone mass from the weight of the forelimbs and hindlimbs.

#### 4.5.2. Forelimb Grip Strength Measurement

Grip strength was measured on day 0 (before the first oral administration), 8, 15, 22, and 29. The tensile force generated by each mouse was measured using a force transducer with a rectangular 4 × 5 cm^2^ metal net (Model-RX-10 Aikoh Engineering Co., Osaka, Japan). Briefly, the mouse was slowly pulled by the tail in the opposite direction while its two forelimbs gripped the metal net. The grip strength was recorded in Newtons (N). Each mouse was subjected to a grip test three times, with at least a 1-min rest between trials, and the maximum grip force was recorded as the forelimb grip strength.

#### 4.5.3. Analysis of Blood Glucose, Cholesterol, Triglyceride, and Free Fatty Acids

Total cholesterol, triglyceride, and glucose levels in serum were measured by the SD LipidoCare^®^ system-analyzer (SD Biosensor, Seoul, Korea). The level of free fatty acids in serum was measured by colorimetric assay kits (FFA, #K612-100, Biovision, Milpitas, CA, USA).

#### 4.5.4. Quantitative Reverse Transcription Polymerase Chain Reaction (qRT-PCR)

Total RNA was isolated from both side of forelimb muscles for qRT-PCR analysis of *Mstn* (myostatin), *Fndc5* (fibronectin type III domain containing 5), *PGC-1a* (peroxisome proliferative activated receptor gamma coactivator 1 alpha), and *Ucp1* (uncoupling protein 1) genes using TRIzol reagents (Invitrogen, Waltham, MA, USA) following the manufacturer’s instructions, and the RNA was used to synthesize cDNA using the PrimeScript^TM^ 1st strand cDNA synthesis kit (Takara, Shiga, Japan). Briefly, 1-µg RNA was added to a 9-µL reaction mixture for reverse transcription and placed in a PCR machine at 65 °C for 5 min. After 5 min, the reaction mixture was adjusted to 20 µL by adding 4 µL of 5x PrimeScript buffer, 0.5 µL of RNase inhibitor, 1 µL of PrimeScript RTase, and 4.5 µL of RNase free dH_2_O. Thereafter, the reaction mixture was incubated at 42 °C for 60 min and finished at 95 °C for 5 min.

PCR amplification was carried out using the iCycler Thermal Cycler (Bio-Rad, Hercules, CA, USA). The 25-µL PCR reaction mixture contained 1 µL of cDNA template (150 ng/µL); 1 µL of each primer (100 pM/µL); 1 µL of dNTP mix (consisting of 0.1 mM each of dATP, dCTP, dGTP, and dTTP); 2.5 µL of 10x buffer; 0.5 µL of *pfu* DNA polymerase; and 18 µL of distilled water. The cycling parameters consisted of an initial incubation at 94 °C for 30 sec, 30 cycles of 15-s denaturation at 94 °C, 30-s annealing at 56 °C, and 30-s extension at 72 °C, and a final 3-min extension step at 72 °C to ensure the complete extension of the amplified products. PCR products were subjected to 1.5% agarose gel electrophoresis and analyzed by the Vilber Lourmat Imaging System (Vilber Luourmat, Marne La Vallee, France). Glyceraldehyde-3-phosphate dehydrogenase (*Gapdh*) was used as an internal control to normalize the expression in determining the relative expression of the target genes. The primer sequences used in this study are shown in Table 2.

#### 4.5.5. Statistical Analysis

The Student’s *t*-test was performed to examine the effect of CSE administration on muscle mass and function, blood metabolites parameters, and gene expressions using the Prism6 program (GraphPad, San Diego, CA, USA).

### 4.6. Isolation of Anti-MSTN Inhibitors from CSE

To isolate the compound(s) having MSTN-inhibitory capacity, the solvent partitioning method was used as illustrated in the following diagram (Figure 9). Dry CSE was mixed with chloroform and water (*v/v* = 1:1). The mixture solution (chloroform and water) was added to the separating funnel and vigorously shaken 3 to 4 times. The funnel was allowed to stand for 1 h without disturbance. Once the layers were separated, the layers were carefully collected. This extraction procedure was repeated two more times, and water fractions were pooled and concentrated using a rotary evaporator. The chloroform layer was also concentrated and subjected to solvent partitioning with 90% ethanol and hexane (*v/v* = 1:1). The hexane layer and 90% ethanol layer were separated and concentrated by the same procedure as above. The 90% ethanol layer was subjected to solvent partitioning with chloroform and 60% ethanol (*v/v* = 1:1), and the layers were separated and concentrated. The water, hexane, 60% ethanol, and chloroform layers were examined for their MSTN-inhibitory activities.

The chloroform layer, which showed MSTN-inhibitory capacity, was subjected to solid-phase extraction chromatography with a C18 Sep-Pak cartridge (Waters, Milford, MA, USA). The active layer was loaded into the cartridge and eluted with water and methanol at ratios of 100:0, 80:20, 60:40, 40:60, 20:80, and 0:100. The factions were examined for their MSTN-inhibitory capacities. The 100% methanol fraction, which showed the highest MSTN-inhibitory capacity, was subjected to reverse-phase HPLC (Pursuit XRs 5 and 10 μm, 10 × 250 mm, Semipreparative C_18,_ 3.5 mL/min, and UV detection at 202 and 280 nm). Elution was performed with an isocratic gradient of 80% methanol for 10 min, then 100% methanol for 20 min, and the fractions were examined for their MSTN-inhibitory capacities. For the structural analysis of the separation of compound 1 and compound 2, they were subjected to a 2nd reverse-phase HPLC (Cosmosil, 10 μm, 10 × 250 mm, Semipreparative C_18_, 75% CH_3_CN, 3 mL/min, and UV detection at 210 and 280 nm).

### 4.7. Identification of MSTN Antagonists

Two distinctive peaks with MSTN-inhibitory capacity were identified using the reverse-phase HPLC separation. To identify the molecular structures of the compounds, LC-MS and NMR experiments were carried out. Liquid chromatography-mass spectrometry (LC-MS) was operated with an LTQ XL linear ion trap (Thermo Scientific, Rockford, IL, USA) equipped with an electrospray ionization (ESI) source that was coupled to a rapid separation LC (RSLC; Ultimate 3000, Thermo Scientific) system (ESI-LC-MS). NMR experiments were operated on the Bruker AVANCE HD 800-MHz NMR spectrometer (Bruker, Berlin, Germany) at the Korea Basic Science Institute (KBSI) in Ochang, Korea. NMR spectra were recorded in CD_3_OD as an internal standard (*δ*_H_ 3.31/*δ*_C_ 49.0).

## Figures and Tables

**Figure 1 molecules-26-02676-f001:**
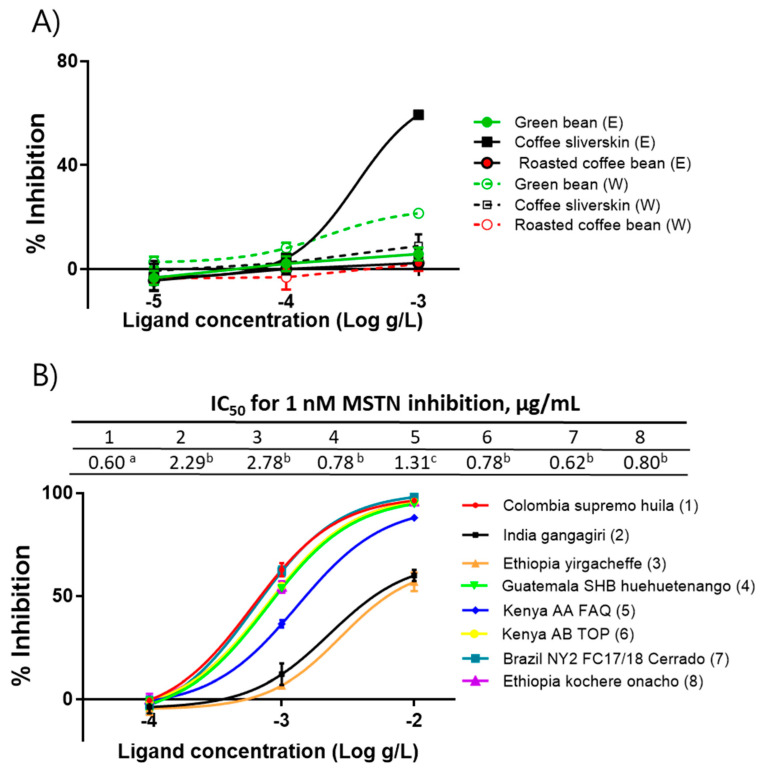
Coffee silverskin ethanol extracts (CSE) inhibit MSTN activity. (**A**) Anti-MSTN activities of the ethanol and water extracts of Green bean, Roasted coffee bean, and Coffee silverskin of Columbia supremo huila. (**B**) Anti-MSTN activities of CSE from various coffee varieties. Various concentrations of CSE in combination with 1-nM MSTN were added to the HEK293 cells transformed permanently with (CAGA)_12_-luciferase gene construct, followed by incubation for 24 h and measurement of the luciferase activity. Percentage of inhibition = (luminescence at 1 nM MSTN—luminescence at each concentration of CSE) × 100/(luminescence at 1 nM MSTN—luminescence at 0 nM MSTN. The error bars indicate SD (*n* = 3). IC_50_ values not sharing the same superscript are different at *p* < 0.05. E, ethanol extract; W, water extract.

**Figure 2 molecules-26-02676-f002:**
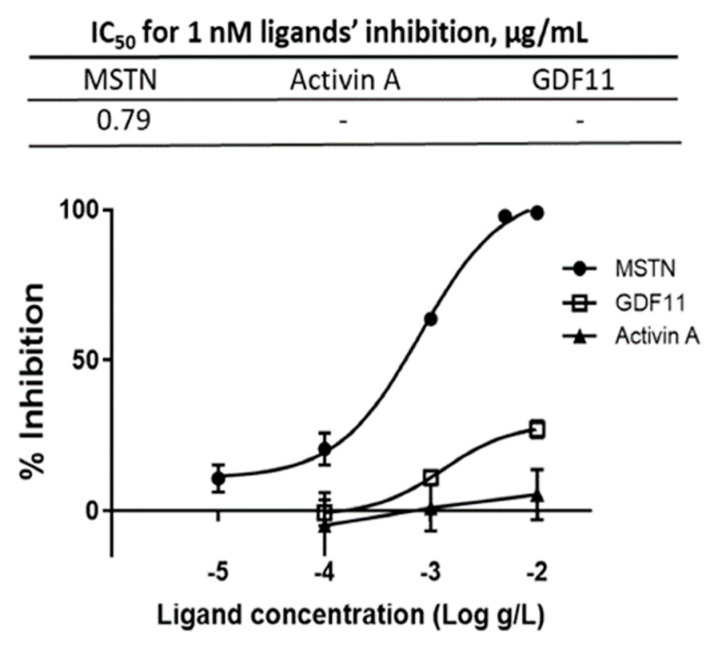
Inhibition of MSTN, Activin A, and GDF11 activities by CSE. Various concentrations of CSE (mixture of various varieties) in combination with 1 nM MSTN, Activin A, or GDF11 were added to the HEK293 cells transformed permanently with (CAGA)_12_-luciferase gene construct, followed by incubation for 24 h and measurement of the luciferase activity. The error bars indicated SD (*n* = 3). Percentage of inhibition = (luminescence at 1 nM ligands—luminescence at each concentration of CSE) × 100/(luminescence at 1 nM ligands—luminescence at 0-nM ligands.

**Figure 3 molecules-26-02676-f003:**
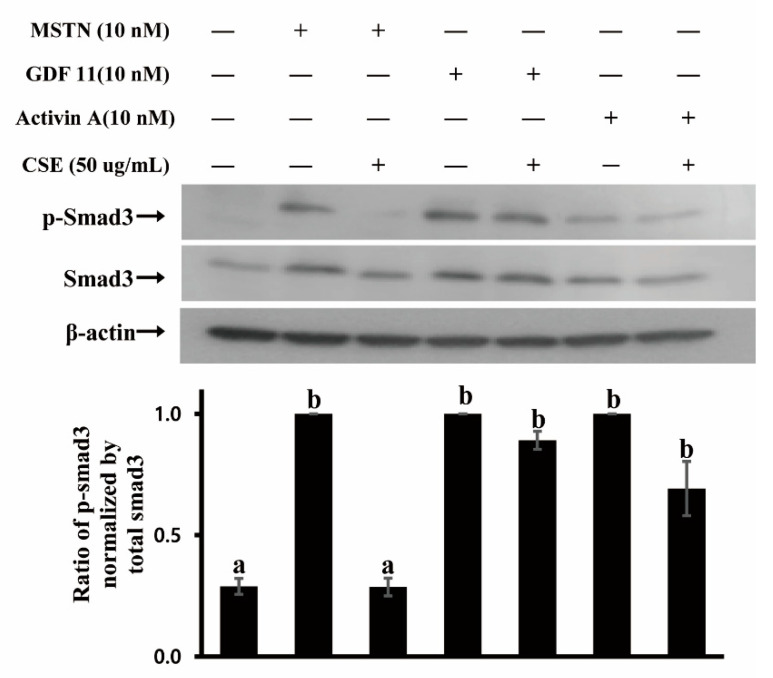
CSE (mixture of various varieties) blocks Smad3 phosphorylation induced by MSTN but not by GDF11 or Activin A in HepG2 cells. Cells were cultured in serum-free DMEM for 4 h, followed by treatment with 10 nM MSTN, 10 nM MSTN plus CSE (50 µg/mL), 10 nM GDF11, 10 nM GDF11 plus CSE (50 µg /mL), 10 nM Activin A, and 10 nM Activin A plus CSE (50 µg /mL) for 30 min. The blots are representative of three independent assays (Appendix A) and have been sequentially probed with antibodies against phospho-Smad3 (p-Smad3), total Smad3 (Smad3), and β-actin. Densitometry analyses of the blots were from three independent assays. The relative phosphorylation level of p-Smad3 was normalized by the expression level of total Smad3. The error bars indicate standard deviation (*n* = 3). Different letters are different at *p* < 0.05.

**Figure 4 molecules-26-02676-f004:**
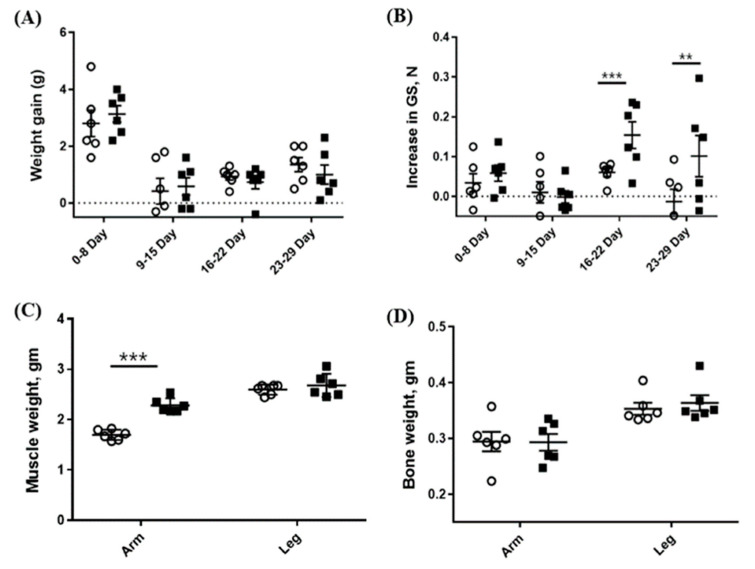
Effects of the oral administration of CSE (mixture of various varieties) on weight gain (**A**), grip strength (**B**), muscle weight (**C**), and bone weight (**D**). Open circle (o) and closed square (■) indicate 0 and 5-mg/kg body weight oral administration of CSE, respectively. The weights of muscle and bone are combined weights of both right and left sides. Data are means ± SEM (*n* = 6). ** *p* < 0.05; *** *p* < 0.001.

**Figure 5 molecules-26-02676-f005:**
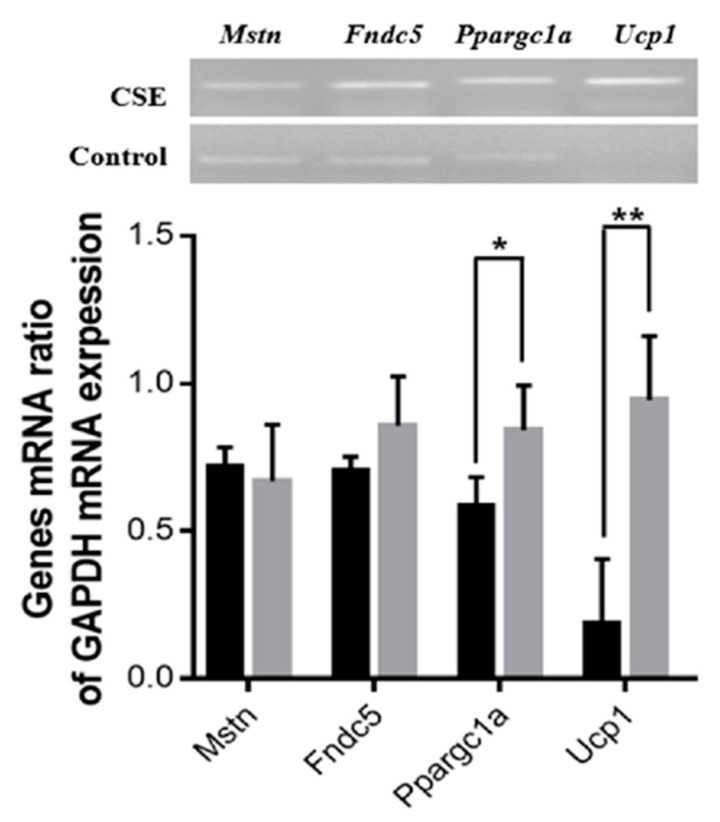
Effects of the oral administration of CSE on the expression of genes related to the metabolism of muscle and fat. The black and gray bars indicate control and CSE-treated groups, respectively. The blots are representative of three independent assays (Appendix A). Data are means ± SE (*n* = 3). Statistical analyses were performed using Student’s *t*-test. * *p* < 0.05; ** *p* < 0.01.

**Figure 6 molecules-26-02676-f006:**
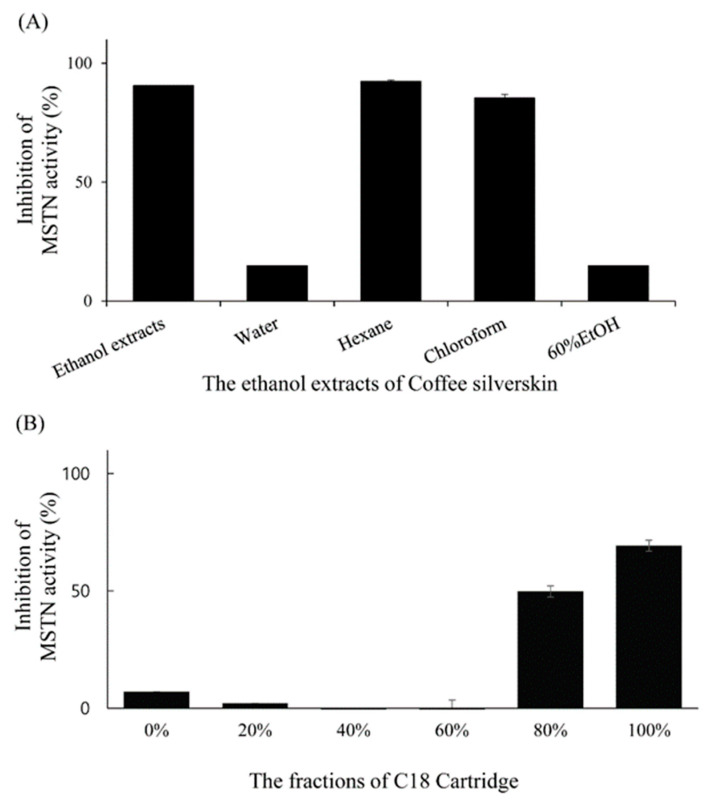
Inhibition of MSTN activity by the organic solvent fractions (Water, Hexane, Chloroform, and 60% Ethanol fractions) of the CSE (**A**) and by the fractions (0%, 20%, 40%, 60%, 80%, and 100% methanol) obtained by applying chloroform layer to C18 Sep-Pak cartridge (**B**). The fractions in combination with 1-nM MSTN were added to the HEK293 cells transformed permanently with (CAGA)_12_-luciferase gene construct, followed by incubation for 24 h and measurement of the luciferase activity. The error bars indicate the SD (*n* = 3).

**Figure 7 molecules-26-02676-f007:**
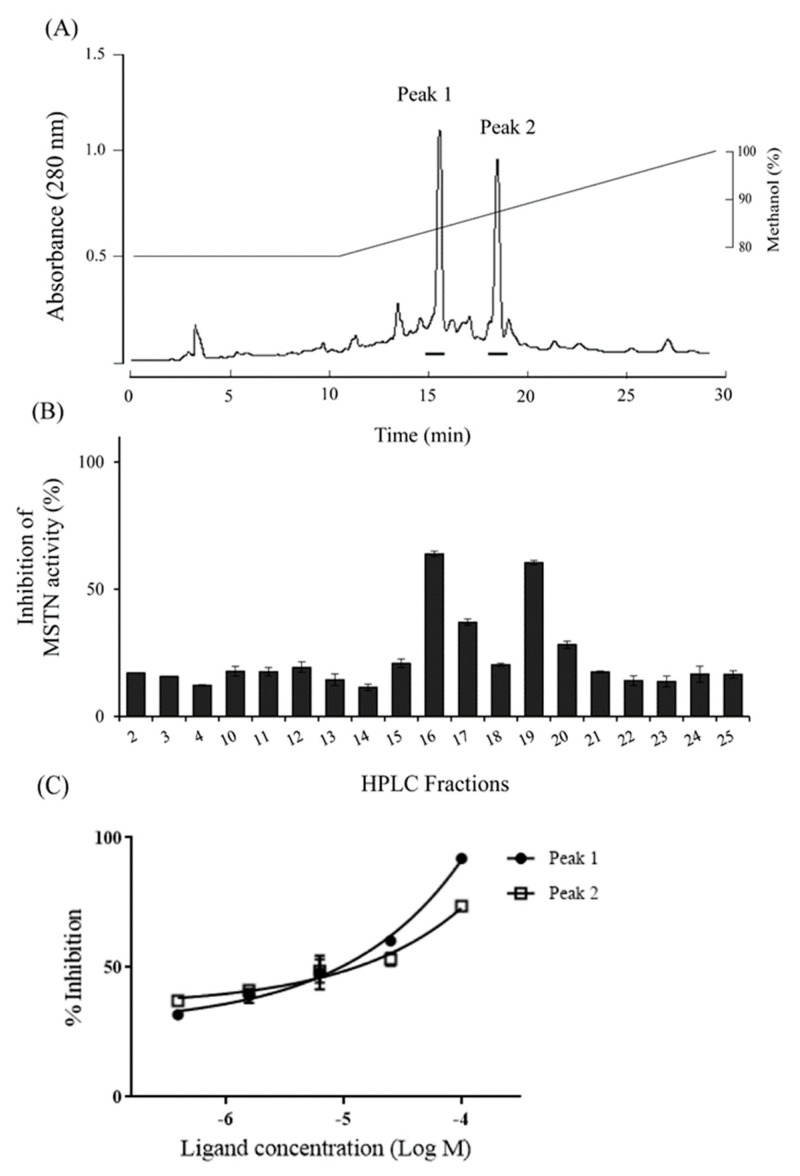
The HPLC profile for the purification of MSTN inhibitors. (**A**) 100% methanol fraction of C18 Sep-Pak cartridge was subjected to the reverse-phase HPLC on a Pursuit XRs 5 C_18_ column (10 × 250 mm, Agilent technology). It was eluted with an isocratic gradient of 80% methanol for 10 min, then 100% methanol for 20 min at a flow rate of 3.5 mL/min. The absorbance was monitored at 280 nm. Bars indicate inhibition parts against MSTN activity. (**B**) Each fraction (1 µg/mL) eluted from HPLC was examined for its inhibitory capacity of 1-nM MSTN. (**C**) Fraction 15 (peak1) and 19 (peak2) at various concentrations were examined for their inhibitory capacities of 1-nM MSTN.

**Figure 8 molecules-26-02676-f008:**
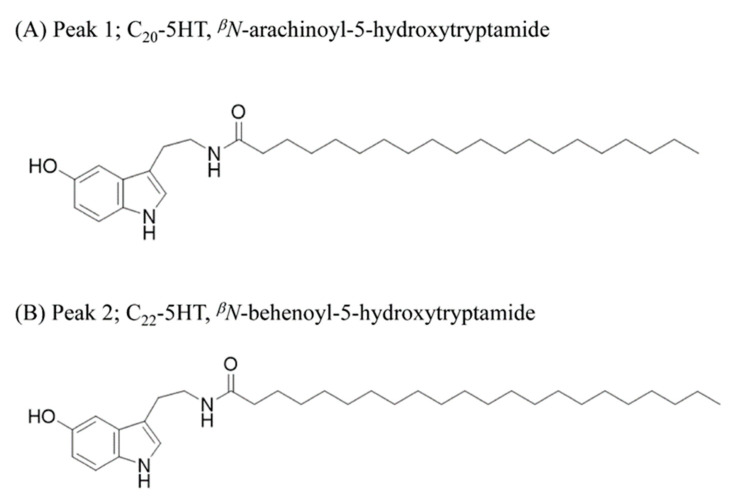
The structures of C_20_-5HT (peak 1) (**A**) and C_22_-5HT (peak 2) (**B**) identified from CSE as MSTN antagonists.

**Figure 9 molecules-26-02676-f009:**
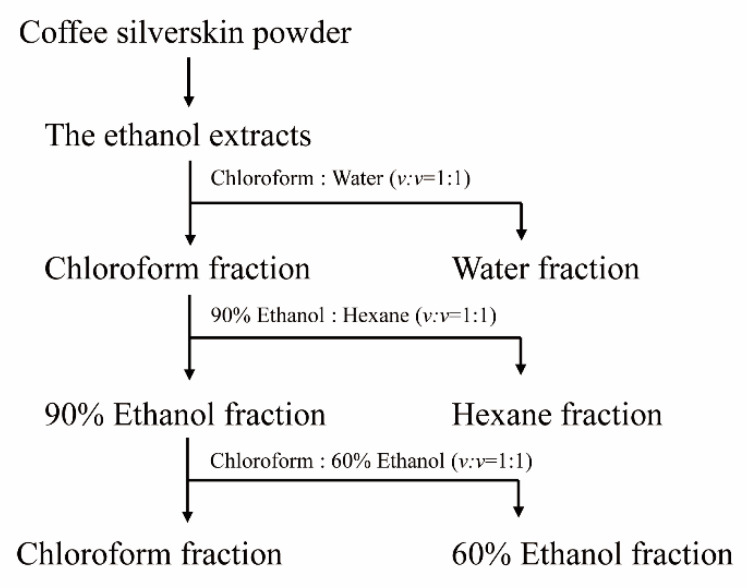
The isolation scheme of CSE using organic solvents.

**Table 1 molecules-26-02676-t001:** Effects of CSE on blood glucose, total cholesterol, triglyceride, and free fatty acid level on day 30.

Group	Glucose (mg/dL)	Total Cholesterol (mg/dL)	Triglyceride (mg/dL)	Free Fatty Acid (µEq/L)
Control	286.0 ± 8.07	127.2 ± 5.74	129.8 ± 11.21	745.8 ± 119.59
CSE	291.8 ± 13.48	132.5 ± 8.12	129.3 ± 11.58	588.6 ± 127.37 ^+^
Relative level (%)	102.0	104.1	99.6	78.9

The relative level is expressed as percentages of the values of the CSE group from those in the control group. Values are means ± SE (*n* = 6). Statistical analyses were performed using Student’s *t*-test. + *p* < 0.1. CSE; The ethanol extracts of coffee silverskin. The relative level is expressed as percentages of the values of the CSE group from those in the control group. Values are means ± SE (*n* = 6). Statistical analyses were performed using Student’s *t*-test. + *p* < 0.1. CSE; the ethanol extracts of coffee silverskin.

**Table 2 molecules-26-02676-t002:** The sequence of primers used for RT-qPCR in this study.

Gene	Forward Primer (5′–3′)	Reverse Primer (5´–3´)	Size (bp)	Accession
*Mstn*	GCTTGACTGCGATGAGCACT	GCACCCACAGCGGTCTACTA	316	NM 010834
*Fndc5*	TTGATGTGGCAGCAGACTTC	CTTATCCGTCCCTCCTCTCC	312	NM_027402
*Ppargc1a*	ATGTGTCGCCTTCTTGCTCT	GCGGTATTCATCCCTCTTGA	350	NM_008904
*Ucp1*	CTTCTCAGCCGGAGTTTCAG	TGCCACACCTCCAGTCATTA	331	NM_009463
*Gapdh*	CGTCCCGTAGACAAAATGGT	TCTCCATGGTGGTGAAGACA	328	NM_001289726

## Data Availability

Not applicable.

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
