# Peer review of "Identification of Molecules from Coffee Silverskin That Suppresses Myostatin Activity and Improves Muscle Mass and Strength in Mice"

_molecules, 2021, doi:10.3390/molecules26092676_

Round 1

Reviewer 1 Report

I found Kim and colleagues' manuscript particularly interesting. The authors investigate the effects of coffee silverskin on muscle function.

In particular, the authors evaluated the hydroalcoholic extract from an analytical point of view and on in vivo models. The authors focus on anti-myostatin activity by evaluating the expression of m-RNA and proteins of the markers involved in the pathways of muscle mass genesis.

The manuscript is clear, the initial theses are well presented and the results discussed. This manuscript enhances the theme of the recovery of waste materials from coffee processing: an added value to the work of Kim and colleagues.
My only suggestion concerns the motor behavior of animals. Did the authors evaluate the motor performance of the animals with the classic tests?

Author Response

Point 1: My only suggestion concerns the motor behavior of animals. Did the authors evaluate the motor performance of the animals with the classic tests?

Response 1 : We could not do the motor behavior test such as the rotarod test. To reflect the lack of motor behavior test, we replace ‘function’ with ‘strength’ in the title of this manuscript and throughout the manuscript.

Reviewer 2 Report

In the manuscript entitled ‘Identification of molecules from coffee silverskin that suppresses myostatin activity and improving muscle mass and function in mice’ authors have investigated the myostatin (MSTN) inhibitory potency of coffee extracts. The ethanol-extract of coffee silverskin has been demonstrated to have anti-MSTN activity and blocked Smad3 phosphorylation induced by MSTN. Further, oral administration of extract enhanced the forelimb muscle mass and grip strength in mice. The manuscript is interesting and obtained results are promising. I have following minor concerns:

  1. Author mentioned that 20 g powder samples were mixed with 1 L of 100% ethanol and incubated at room temperature under a dark condition for 24 h. Why incubated in dark for 24h, if any reference study is there then it should be cited.
  2. If author has used the 1 μg RNA for cDNA synthesis it means author has already normalized the RNA for all the experiment, then why 150 ng of cDNA template was used for PCR amplification.
  3. Abbreviations throughout the manuscript should be checked and corrected.

Author Response

Point 1: Author mentioned that 20 g powder samples were mixed with 1 L of 100% ethanol and incubated at room temperature under a dark condition for 24 h. Why incubated in dark for 24h, if any reference study is there then it should be cited.

Response 1: When we prepared an extract from a sample, we extracted it in the dark to reduce light-induced chemical reactions. The section is revised and read as follows: “------samples were mixed with 1 L of 100% ethanol and incubated at room temperature for 24 h under a dark condition to reduce light-induced chemical reactions.”

Point 2: If author has used the 1 μg RNA for cDNA synthesis it means author has already normalized the RNA for all the experiment, then why 150 ng of cDNA template was used for PCR amplification.

Response 2: It is correct that the RNA concentration was normalized by using 1 μg of RNA. A fixed volume (1 μL) was used from the reverse transcription reaction mixture. The 1 μL was equivalent to 150 ng of cDNA. We revised this part and it reads as follows: ‘PCR amplification was carried out using the iCycler thermal cycler (Bio-rad, CA, USA). The 25 µl PCR reaction mixture contained 1 µl of cDNA template (150 ng/µl), 1 µl of each primer (100 pM/µl), 1 µl of dNTP mix (consisting of 0.1 mM each dATP, dCTP, dGTP, and dTTP), 2.5 µl of 10x buffer, 0.5 µl of pfu DNA polymerase, and 18 µl of distilled water. The cycling parameters consisted of an initial incubation at 94oC for 30 sec; 30 cycles of 15 sec denaturation at 94 oC, 30 sec annealing at 56 oC, and 30 sec extension at 72 oC; and a final 3 min extension step at 72 oC to ensure the complete extension of the amplified products.”

Point 3: Abbreviations throughout the manuscript should be checked and corrected.

Response 3: We checked and corrected abbreviations in the manuscript.

Reviewer 3 Report

The paper of Kim et al. investigates the possibility that different coffee bean components may modulate the activation of TGFβ signalling in skeletal muscle. Myostatin (Mstn) is one of the major TGFβ factor in skeletal muscle and it acts by negatively controlling muscle mass especially in postnatal tissue. The authors analysed the capacity of green bean, roasted bean and silver skin extracts to inhibit the Myostatin pathway and thus exerting a hypertrophic effect on skeletal muscle. Silver skin ethanol extracts (CSE) showed the highest inhibitory effect on Mstn-dependent transcription. Of different coffee bean varieties, the authors also identified those with the highest impact on Mstn inhibition (in order: Colombia supremo huila, Guatemala SHB huehuetenango, Kenya AB TOP, Brazil NY2FC17/18 Cerrado). They also tested the in vivo efficacy of CSE by administering it to experimental animals for 29 days and testing the effect of the treatment on muscle mass and muscle strength.

The findings are quite interesting and promising, leading to the identification of new natural compounds which are able to modulate Mstn signaling and thus to induce muscle hypertrophy. However, the paper has some points of weakness and, at the present state, it might be not ready for publication. The major flaw of the work is the lack of investigation about muscle hypertrophy. The effect of CSE on muscle mass is central for the meaningfulness of the article, despite that it is not considered at the histological and molecular level by the authors. No single analysis of fiber size and anabolic pathways (Akt, protein synthesis, etc.) has been provided. In addition, I highlighted here as follows, some of the main issues that should be addressed in order to be able to reconsider the manuscript.

Major issues:

  1. The authors performed the analysis of the Mstn inhibitory capacity of the CSE from different coffee varieties in Figure 1B. However, they do not state which variety was used for the rest of the experiments in the following experimental work. Did they select the most effective CSE (from Colombia supremo huila) or the CSE used came from a mixture of different coffee sources?
  2. Figure 1: the use of different line markers to identify samples is not ideal since the line markers overlap and are not clearly recognized in the graph. Rather use different line colours.
  3. Did the author quantify the presence of the N-alkanoyl-5-hydroxytryptamides (C20-5-HT and C22-5-HT) in the extracts of green bean, roasted bean and water-silver skin extracts that do not show Mstn inhibitory activity, in order to prove that are indeed those molecules performing the negative action on the Mstn pathway and that they are present in high quantity exclusively in silverskin?
  4. Why did the author decide to test the activity of CSE on TGFβ factors on HEK293 cells (Figure 2) rather than on myogenic cells, such as muscle-derived primary cells or the commonly used C2C12 myoblast cell line? This option will be informative on the effect of CSE on Mstn transduction pathway in a system that more closely resemble the skeletal muscle tissue, and it would have been useful for additional biochemical studies (such as biochemical analysis of the Akt pathway activation, see my general comment and the following points 6 and 7).
  5. The authors used 1 nM Mstn for 24h to trigger Luciferase reporter gene transcription in HEK293 cells (Figure 2) while they increase the concentration to 10 nM for the analysis of Smad3 phosphorylation (Figure 3). I understand the Mstn concentration for the treatments should be adjusted and may vary in long- versus short-term treatments (i.e. 24h vs 30min); however, the authors should comment and explain the choice of the two different concentrations in the text.
  6. The assessment of muscle mass is quite approximate. Besides the measure of the weight of muscles (which muscles? The anatomical identification of each muscle should be provided), the authors did not analysed fiber size nor fiber number or the number of nuclei per fiber. Those parameters are crucial to define the occurrence of muscle hypertrophy rather than hyperplasia.
  7. The effect of CSE on muscle mass is one of the central result of the paper but it remains unfortunately underinvestigated. A deeper insight on the role of CSE on Mstn inhibition should have been accomplished. For example, the assessment of Smad3 phosphorylation in the hypertrophic muscles of treated animals would have confirmed the effective inhibition of Mstn signalling by CSE. Moreover, the analysis of Akt/S6K pathway activation would clarify if this pathway, already known to have a major contribution to muscle hypertrophy also in response to Msnt downregulation (Trendelenburg AU, Meyer A, Rohner D, Boyle J, Hatakeyama S, Glass DJ (2009) Myostatin reduces Akt/TORC1/p70S6K signaling, inhibiting myoblast differentiation and myotube size. Am J Physiol Cell Physiol 296(6):C1258–C1270. doi:10.1152/ ajpcell.00105.2009), is involved in the effect of CSE.
  8. Finally, the fact that hindlimb muscles do not seem affected by CSE is definitively surprisingly. The authors just mention the observation without further comment and discuss this unexpected finding. An extensive comparison between responding (forelimbs ) versus non responding (hindlimb) muscles should have been accomplished in terms of muscle mass, fiber size, nuclear content and molecular pathway analysis (both transcriptional and biochemical), in order to explain the different outcomes obtained in the two muscle groups.
  9. In the same line, the finding that only forelimbs but not hindlimbs undergo muscle hypertrophy is difficult to explain and the hypothesis about different behaviour of muscles of the same treated animal is hazardous. Maybe the measure of muscle weight is not accurate enough to reveal the increase of muscle mass in the posterior muscles. More detailed analysis of hindlimb muscles should have been performed to definitively exclude their response to CSE treatment. Do the author measure fiber size in those muscles? Do they assess muscle force in the hindlimbs? Those measurements should be provided.
  10. The authors reported no changes in body weight, despite a significant increase in muscle mass (at least in the forelimbs, Figure 4C). Considering that 60-70% of mouse body weight might consist of muscle mass (G. E. GRIFFIN AND G. GOLDSPINK. The Increase in Skeletal Muscle Mass in Male and Female Mice 1973. ANAT. REC., 177: 465470). How is it conceivable that muscle hypertrophy induced by CSE-mediated Mstn inhibition does not lead to increased body weight?

Minor issues:

  • Line 43 is repeated and should be deleted
  • The details of qRT-PCR should be provided: how many cycles of amplification did the authors performed (lines 401-402)? Moreover, the Biorad iCycler is supposed to be a real time PCR instrument, why authors did run an electrophoresis gel after the PCR amplification? The graph in Figure 5 related also to Figure S2 represent the real time PCR data or the electrophoresis gel data? This issue should be clarified.
  • The images of PCR product in Figure S2 are very faint, especially that of the 3rd Better quality/contrasted panels should be provided.

Author Response

Point 1: The authors performed the analysis of the Mstn inhibitory capacity of the CSE from different coffee varieties in Figure 1B. However, they do not state which variety was used for the rest of the experiments in the following experimental work. Did they select the most effective CSE (from Colombia supremo huila) or the CSE used came from a mixture of different coffee sources?

Response 1: In subsequent experiments, CSEs from all varieties of coffee silverskin were mixed and used, and this is noted in figure legends and the Materials and Methods section.

Point 2: Figure 1: the use of different line markers to identify samples is not ideal since the line markers overlap and are not clearly recognized in the graph. Rather use different line colours.

 Response 2: As suggested, different colors were used

Point 3: Did the author quantify the presence of the N-alkanoyl-5-hydroxytryptamides (C20-5-HT and C22-5-HT) in the extracts of green bean, roasted bean and water-silver skin extracts that do not show Mstn inhibitory activity, in order to prove that are indeed those molecules performing the negative action on the Mstn pathway and that they are present in high quantity exclusively in silverskin?

Response 3: We agree with the comment. However, at the time of study, we focused on identifying compounds having MSTN-inhibitory capacity. We believe future studies need to examine the presence of C20-5-HT and C22-5-HT in other extracts.

Point 4: Why did the author decide to test the activity of CSE on TGFβ factors on HEK293 cells (Figure 2) rather than on myogenic cells, such as muscle-derived primary cells or the commonly used C2C12 myoblast cell line? This option will be informative on the effect of CSE on Mstn transduction pathway in a system that more closely resemble the skeletal muscle tissue, and it would have been useful for additional biochemical studies (such as biochemical analysis of the Akt pathway activation, see my general comment and the following points 6 and 7).

Response 4: The HEK293 is a transformed cell stably expressing pGL-(CAGA)12-luciferase gene construct, which expresses luciferase in response to MSTN-binding to its receptor. The system is very sensitive and routinely used for the measurement of MSTN or MSTN-inhibitory activity, as well as activities of other TGF-beta family member proteins. Using C2C12 requires stable transformation with pGL-(CAGA)12-luciferase gene construct requiring time-consuming process, while the PI’s lab has the HEK293 cells stably expressing pGL-(CAGA)12-luciferase gene construct.     

Point 5: The authors used 1 nM Mstn for 24h to trigger Luciferase reporter gene transcription in HEK293 cells (Figure 2) while they increase the concentration to 10 nM for the analysis of Smad3 phosphorylation (Figure 3). I understand the Mstn concentration for the treatments should be adjusted and may vary in long- versus short-term treatments (i.e. 24h vs 30min); however, the authors should comment and explain the choice of the two different concentrations in the text.

Response 5: The pGL-(CAGA)12-luciferase reporter gene assay is designed to examine the MSTN-inhibitory activity. The strength of luciferase activity is proportional to the concentration of MSTN to a certain level, and luciferase activity gradually increases and plateau at around 8 h and maintain the activity up to 24 h. 1 nM has been shown to be enough to generate strong luciferase activity, so we have been using the 1 nM concentration and 24 h incubation.

Smad3 phosphorylation with HepG2 cells was based on previous studies (a reference is cited), and we found that 10 nM of MSTN with 20 min incubation was an appropriate condition for Smad3 phosphorylation.

We used different cell types for each assay and found that the appropriate MSTN concentrations and incubation time were different between the two assays.

Point 6: The assessment of muscle mass is quite approximate. Besides the measure of the weight of muscles (which muscles? The anatomical identification of each muscle should be provided), the authors did not analysed fiber size nor fiber number or the number of nuclei per fiber. Those parameters are crucial to define the occurrence of muscle hypertrophy rather than hyperplasia.

Response 6: We agree with the comment. The muscle weight measurement was a bit crude and measurement of fiber size and number would provide more and deeper insights into the effects of CSE on muscle mass. However, due to unforeseen limitations, we could not perform histological analysis of muscle fibers. Nevertheless, we believe that the total forelimb and hindlimb muscle wt measurement provide unbiased information regarding the effect of CSE on muscle mass even though the underlying mechanism cannot be explained. We added a statement regarding the limitation of the result in the discussion section.

Point 7: The effect of CSE on muscle mass is one of the central result of the paper but it remains unfortunately underinvestigated. A deeper insight on the role of CSE on Mstn inhibition should have been accomplished. For example, the assessment of Smad3 phosphorylation in the hypertrophic muscles of treated animals would have confirmed the effective inhibition of Mstn signalling by CSE. Moreover, the analysis of Akt/S6K pathway activation would clarify if this pathway, already known to have a major contribution to muscle hypertrophy also in response to Msnt downregulation (Trendelenburg AU, Meyer A, Rohner D, Boyle J, Hatakeyama S, Glass DJ (2009) Myostatin reduces Akt/TORC1/p70S6K signaling, inhibiting myoblast differentiation and myotube size. Am J Physiol Cell Physiol 296(6):C1258–C1270. doi:10.1152/ ajpcell.00105.2009), is involved in the effect of CSE.

Response 7: We agree that the suggested mechanistic investigation would have brought valuable insights into the effects of CSE on muscle protein metabolism in association with MSTN. However, again, at the time of this study, we could not perform the mechanistic investigation due to certain limitations at the time of the study. Currently, a study is underway to investigate a wide range of questions in association with the role of CSE and/or C20-5-HT and C22-5-HT on muscle mass and function and their underlying mechanisms. We briefly added the limitation of the result in the discussion section: “This study, however, could not thoroughly examine underlying signaling pathways associated with muscle protein metabolism regulated by MSTN, as well as muscle cellularities such as muscle fiber hypertrophy and hyperplasia, and fiber type composition, thus future studies need to investigate the underlying mechanisms.”  

Point 8: Finally, the fact that hindlimb muscles do not seem affected by CSE is definitively surprisingly. The authors just mention the observation without further comment and discuss this unexpected finding. An extensive comparison between responding (forelimbs) versus non responding (hindlimb) muscles should have been accomplished in terms of muscle mass, fiber size, nuclear content and molecular pathway analysis (both transcriptional and biochemical), in order to explain the different outcomes obtained in the two muscle groups.

Response 8: The data was truly unexpected. We admit that no reasonable explanation is possible with the current data. However, we could not find any errors related to sample collection and data processing. Even though unexpected, that was the data collected from the study, and we were compelled to accept and report. The following sentences are added in the discussion section to describe the unexpected nature of the result: “Unexpectedly, hindlimb muscle mass was not affected by the CSE administration. With the current data, the reason for the differential effect cannot be explained, and future studies are needed to confirm and explain the result with detailed mechanistic investigations”

Point 9: In the same line, the finding that only forelimbs but not hindlimbs undergo muscle hypertrophy is difficult to explain and the hypothesis about different behaviour of muscles of the same treated animal is hazardous. Maybe the measure of muscle weight is not accurate enough to reveal the increase of muscle mass in the posterior muscles. More detailed analysis of hindlimb muscles should have been performed to definitively exclude their response to CSE treatment. Do the author measure fiber size in those muscles? Do they assess muscle force in the hindlimbs? Those measurements should be provided.

Response 9: Unfortunately, we could not measure the size of the muscle fibers and the strength of the hindlimbs. The above statement we added addresses the limitation. 

Point 10: The authors reported no changes in body weight, despite a significant increase in muscle mass (at least in the forelimbs, Figure 4C). Considering that 60-70% of mouse body weight might consist of muscle mass (G. E. GRIFFIN AND G. GOLDSPINK. The Increase in Skeletal Muscle Mass in Male and Female Mice 1973. ANAT. REC., 177: 465470). How is it conceivable that muscle hypertrophy induced by CSE-mediated Mstn inhibition does not lead to increased body weight?

Response 10: Unfortunately, we did not measure fat content. It is speculated that fat mass was reduced. Preliminary results from a recent study with an isolated substance indicate that fat mass is decreased by the substance.

The following statement is added in the discussion section: “Notably, despite the increase in forelimb muscle mass, body weight was not affected by the administration of CSE. The fat content was not measured in this experiment. Given that MSTN inhibition is known to reduce fat accumulation in mice [17, 18], it is likely that the reason for no change in body weight was due to a decrease in fat mass.”

Point 11: Line 43 is repeated and should be deleted:

Response 11: Deleted

Point 12: The details of qRT-PCR should be provided: how many cycles of amplification did the authors performed (lines 401-402)? Moreover, the Biorad iCycler is supposed to be a real time PCR instrument, why authors did run an electrophoresis gel after the PCR amplification? The graph in Figure 5 related also to Figure S2 represent the real time PCR data or the electrophoresis gel data? This issue should be clarified.

Response 12: The iCycler Thermal Cycler cannot be used for real-time PCR without an optical module, and our cycler was not equipped with an optical module.  

Point 13: The images of PCR product in Figure S2 are very faint, especially that of the 3rd Better quality/contrasted panels should be provided.

Response 13:  A better panel for #3 is added.

Round 2

Reviewer 3 Report

I carefully read the authors point-by-point response and I appreciated the completeness of their reply. Despite not all my suggestions and requests were accomplished and satisfied, the authors carefully answered to each one of my comments, giving explanations and eventually providing pieces of information on the issues I raised. Nevertheless, some of my major comments (point 6, 7, 9 and 10 are the more crucial)  were not addressed due to not better specified “limitations at the time of the study”. This unfortunately add a degree of weakness to the work.

Concluding, the article of Kim et al. did not acquire the desired soundness that it might have deserved since the authors finally did not implement any new experimental findings to the original data. However, the paper has been certainly ameliorated after the revision and the new manuscript version has a clearer presentation of the data and a more detailed discussion of the results.